# Comparing patients' and other stakeholders' preferences for outcomes of integrated care for multimorbidity: a discrete choice experiment in eight European countries

Maureen Rutten-van Mölken ,[1] Milad Karimi,[1] Fenna Leijten,[1,2] Maaike Hoedemakers,[1] Willemijn Looman,[1] Kamrul Islam,[3] Jan E Askildsen,[3] Markus Kraus,[4] Darija Ercevic,[5] Verena Struckmann,[6] János Gyorgy Pitter,[7] Isaac Cano,[8] Jonathan Stokes,[9] Marcel Jonker,[1] On behalf of the SELFIE consortium

For numbered affiliations see end of article.

**Correspondence to**
Dr Maureen Rutten-van Mölken;
m.rutten@eshpm.eur.nl

## ABSTRACT

**Objectives** To measure relative preferences for outcomes of integrated care of patients with multimorbidity from eight European countries and compare them to the preferences of other stakeholders within these countries.

**Design** A discrete choice experiment (DCE) was conducted in each country, asking respondents to choose between two integrated care programmes for persons with multimorbidity.

**Setting** Preference data collected in Austria (AT), Croatia (HR), Germany (DE), Hungary (HU), the Netherlands (NL), Norway (NO), Spain (ES), and UK.

**Participants** Patients with multimorbidity, partners and other informal caregivers, professionals, payers and policymakers.

**Main outcome measures** Preferences of participants regarding outcomes of integrated care described as health/well-being, experience with care and cost outcomes, that is, physical functioning, psychological well-being, social relationships and participation, enjoyment of life, resilience, person-centredness, continuity of care and total costs. Each outcome had three levels of performance.

**Results** 5122 respondents completed the DCE. In all countries, patients with multimorbidity, as well as most other stakeholder groups, assigned the (second) highest preference to enjoyment of life. The patients top-three most frequently included physical functioning, psychological well-being and continuity of care. Continuity of care also entered the top-three of professionals, payers and policymakers in four countries (AT, DE, HR and HU). Of the five stakeholder groups, preferences of professionals differed most often from preferences of patients. Professionals assigned lower weights to physical functioning in AT, DE, ES, NL and NO and higher weights to person-centredness in AT, DE, ES and HU. Payers and policymakers assigned higher weights than patients to costs, but these weights were relatively low.

**Conclusion** The well-being outcome enjoyment of life is the most important outcome of integrated care in multimorbidity. This calls for a greater involvement of

## Strengths and limitations of this study

► We designed an online discrete choice experiment (DCE) to measure the relative importance of health, well-being, experience and cost outcomes of integrated care to patients with multimorbidity from eight European countries.

► The choice of outcomes was largely driven by focus groups involving patients with multimorbidity.

► We compared patients' preferences with the preferences of other stakeholders including informal caregivers, professionals, payers and policymakers in a large sample of 5122 respondents.

► Online discrete choice experiments are cognitively challenging and respondents may not represent all patients with multimorbidity.

► When completing the DCE, it might be challenging for the other stakeholders to separate their professional role from the other roles they might have.

social and mental care providers. The difference in opinion between patients and professionals calls for shared decision-making, whereby efforts to improve well-being and person-centredness should not divert attention from improving physical functioning.

## INTRODUCTION

Persons with multimorbidity may have complex care needs requiring the involvement of different types of professionals from different sectors of the healthcare, social care and/or community care systems. If the services provided by different professionals are not well aligned, they may experience conflicting treatment goals, harmful effects of interacting treatments, overly-demanding

appeals on self-management capabilities and inefficiencies like overtreatment and undertreatment.

It is hypothesised, that persons with multimorbidity would benefit from person-centred integrated care programmes.[1–3] Integrated care programmes commonly adopt a holistic approach towards people with multimorbidity, which means that, instead of the disease, the person with his or her needs, capabilities, preferences, resources and (social) environment is put in the centre.[4] In the European Union (EU)-funded Horizon2020 project SELFIE (**S**ustainable int**E**grated chronic care mode**L**s for multimorbidity: delivery, **FI**nancing, and performanc**E**), we study 17 promising integrated care programmes from eight European countries, that is, Austria (AT), Croatia (HR), Germany (DE), Hungary (HU), the Netherlands (NL), Norway (NO), Spain (ES) and UK. In many of these programmes an interdisciplinary individual care plan, with well-aligned health and social care services, is designed to specifically target the goals and priorities jointly set by the person with multimorbidity and his or her formal and informal caregivers. Many programmes also offer tailored self-management support and assign a case manager to the most complex patients, to ensure that they have a single contact point that helps them navigate through the system.

Integrated care programmes for persons with multimorbidity often aim to improve outcomes that go beyond health and include their broader sense of well-being, experience with care and costs. This is often referred to as the Triple Aim of improving population health/well-being, experience and costs.[5] Therefore, we decided to measure a broad set of outcome measures in the evaluation studies of the 17 programmes in the SELFIE project. However, the importance of these outcomes is almost never measured. In the SELFIE project we were specifically interested in the importance of these outcomes to persons with multimorbidity. Understanding their preferences is vital for the optimal design of person-centred integrated care services and inform decision-making on implementation and upscaling of these services. In the SELFIE project, we aim to use the preferences as weights in Multi-Criteria Decision Analyses (MCDA) of the 17 programmes.[6] In an MCDA, a wide range of different outcomes is aggregated into a single, overall value score. This overall value score combines the performance of the integrated care programmes on different outcomes, each weighted according to its respective preference weight.[7 8] Besides the persons with multimorbidity, there are many other stakeholders involved in decision-making about integrated care and their preferences may differ from patients' preferences. Therefore, we aim to repeat the MCDA's in the SELFIE project, using the preference weights from different groups of stakeholders.

The aim of the current paper is twofold. First, to present the relative magnitude of the preferences for the Triple Aim outcomes in persons with multimorbidity from eight European countries. Second, the aim is to compare the Patients' preferences to the preferences of Partners and other informal caregivers, Professionals, Payers and Policymakers, together referred to as the 5 P's. This will be a within-country comparison, because the decision context of the future MCDA's relates to local or regional level decisions about the adaptation, reimbursement, continuation and scaling-up of the 17 existing integrated care programmes.

## METHODS

### Discrete choice experiment

The preferences were obtained using discrete choice experiments (DCEs). A DCE is a theoretically well founded method to determine how much respondents are willing to trade-off some outcomes against others.[9–11] In our DCE, respondents from the 5P's were given questions that asked them to choose between two integrated care programmes for persons with multimorbidity, labelled programme A and programme B. There was no opt-out. Each programme was described in terms of outcomes with systematically varying levels that describe the performance of the programme on the outcomes. The outcomes were largely based on focus groups and included measures of health and well-being: (1) physical functioning, (2) psychological well-being, (3) social relationships and participation, (4) enjoyment of life, (5) resilience; measures of experience: (6) person-centredness, (7) continuity of care; and (8) total healthcare and social care costs. The definitions are given in table 1. An example of a DCE question is given in figure 1.

Each outcome (attributes in DCE terminology) had three levels indicating a poor, average and good performance of the programme on that outcome. The definitions of the outcomes and levels were identical across the eight countries, except for costs. The three levels of costs were based on country-specific estimates of the mean total health and social care costs for people with multimorbidity in 2017 (middle level) and increased and decreased by 20% to obtain the poor and good performance level. This percentage was chosen to ensure enough variation in levels while at the same time remaining within realistic boundaries. The costs were expressed in the national currency.

### DCE design

A Bayesian efficient heterogeneous design algorithm was used to optimise the DCE designs based on the D-efficiency criterion.[12] Ten different subdesigns with 18 choice tasks per subdesign were used. In each subdesign either four or five attribute levels were overlapped between programme A and B, to reduce the overall complexity of the choice tasks and improve response efficiency.[13] One restriction was built into the design, namely, the best level of the psychological well-being attribute and the worst level of enjoyment of life, and vice versa, were not presented simultaneously for the same programme.

The initial DCE designs were optimised using priors that were based on the available literature (see online

**Table 1** Definitions of the outcomes and levels

| Outcome (attribute) | Definition | Levels |
|---|---|---|
| Physical functioning | Acceptable physical functioning and being able to do daily activities without needing assistance (eg, getting dressed, sitting down and getting up from a chair, taking your medications) | 1. Severely limited in physical functioning and activities of daily living<br>2. Moderately limited in physical functioning and activities of daily living<br>3. Hardly or not at all limited in physical functioning and activities of daily living |
| Psychological well-being | The absence of stress, worrying, listlessness, anxiety and feeling down | 1. Always or mostly stressed, worried, listless, anxious and down<br>2. Regularly stressed, worried, listless, anxious and down<br>3. Seldom or never stressed, worried, listless, anxious and down |
| Social relationships and participation | Having meaningful connections with others as desired | 1. No or barely any meaningful connections with others<br>2. Some meaningful connections with others<br>3. A lot of meaningful connections with others |
| Enjoyment of life | Having pleasure and happiness in life | 1. No or barely any pleasure and happiness in life<br>2. Some pleasure and happiness in life<br>3. A lot of pleasure and happiness in life |
| Resilience | The ability to recover from or adjust to difficulties and to restore one's balance | 1. Poor ability to recover, adjust and restore balance<br>2. Fair ability to recover, adjust and restore balance<br>3. Good ability to recover, adjust and restore balance |
| Person-centredness | Care that matches an individual's needs, capabilities and preferences and where decisions are made jointly based on good information | 1. Not or barely person-centred<br>2. Somewhat person-centred<br>3. Highly person-centred |
| Continuity of care | Good collaboration, smooth transitions between caregivers and no waste of time | 1. Poor collaboration, transitions and timeliness<br>2. Fair collaboration, transitions and timeliness<br>3. Good collaboration, transitions and timeliness |
| Total healthcare and social care costs | Total healthcare and social care costs per participant in the programme, per year. Note: These are costs paid for by the health insurer/government. | 1. 8500 € per participant per year*<br>2. 7000 € per participant per year<br>3. 5500 € per participant per year |

*These are the Dutch values. The value for the other countries are: Austria: 8000, 6600, 5200 €; Croatia: 7200, 6000, 4800 kuna (973, 810, 648 €); Hungary: 600 000, 500 000, 400 000 forint (1951, 1626, 1300 €); Germany: 4800, 4000, 3200 €; Norway: 115 000, 95 000, 75 000 krone (12 330, 10 185, 8041 €); Spain: 5400, 4500, 3600 euro; UK: 3600, 3000, 2400 pounds (4130, 3441, 2753 €).
€, Euro.

supplemental box S1). These DCE designs were subsequently updated, based on updated priors that reflected the information obtained from approximately the first 50 respondents in each stakeholder group. The design updates were performed separately for each country/ stakeholder group and the updated priors were the average of the literature-based priors and the conditional logit estimates of the first 50 respondents.

### Questionnaire design
The online DCE survey started with an explanation of the SELFIE project, the outcomes, type of questions and the perspective from which the questions should be answered (ie, one of the five P's). Each respondent completed a randomly chosen subdesign with 18 choice tasks, preceded by two practice-questions. The order of appearance of the choice tasks within each subdesign was randomised. The 18 choice tasks were presented in three sections of six choice tasks each, with several background questions in between. The background questions related to sociodemographic characteristics and the respondents' health conditions. Colour coding was used for the different levels of the outcomes as shown in figure 1, and outcomes that had the same level in the two integrated care programmes were presented in grey.[13] When respondents hovered the mouse over an outcome, the definition of that outcome became visible. At the end of the survey, respondents received a multiple-choice question on the level of difficulty of the questionnaire.

**Figure 1** Example of a discrete choice experiment question.

The questionnaire was designed in English and translated into each country's respective language, using the same translation protocol, which included forward and backward translations by native-language speakers with an excellent level of English.

**Respondents**

Respondents from the stakeholder groups were recruited in the eight countries participating in the SELFIE project during 2017 to 2018. Sample size calculations[14] led to a target number of 150 respondents per country per stakeholder group. Each country recruited respondents from three to five different groups of stakeholders. All countries recruited respondents from the first three P's (ie, patients with multimorbidity, partners and other informal caregivers, professionals). The Netherlands and Norway recruited respondents from all five P's. The other countries combined the respondents from the payers and policymakers into one group, either because they have a Beveridge-like healthcare model in which there is a national/regional health service with a single payer (HR, HU, UK and ES) or because they were in the midst of a comprehensive health insurance reform (AT) or an election (DE), making it difficult to recruit a sufficient number of respondents from the two groups separately.

*Patients with multimorbidity* and *partners and other informal caregivers* were recruited from internet panels managed by market research organisations, except in Spain where they were personally invited to participate by the members of the Spanish team of the SELFIE project who were physicians and nurses. In Spain, patients were recruited in the waiting rooms of a hospital, geriatric residences and support groups for families with Alzheimer. All patients had two or more chronic conditions. Partners and other informal caregivers had to provide voluntary care or support to a family member, friend or other acquaintance who needed help due to physical, mental or ageing-related health problems, for at least 2 weeks.

*Professionals* were also recruited through an Internet panel organisation in the UK and the Netherlands. In the other countries, they were recruited through the networks of the authors and a snowball sampling method, in which we asked respondents if they could invite other responders within their organisations. Examples of organisations approached include care provider organisations or professional associations like nursing associations.

All *payers* and *policymakers* in the UK and about one-third of the payers and one-sixth of the policymakers in the Netherlands were recruited via an Internet panel.

Payers had to have paid employment in an occupation in which they were (in)directly involved in the financing or payment of integrated care and/or care for persons with multimorbidity. Policymakers had to be (in)directly involved in policymaking or decision-making about healthcare, social care or welfare. In the other countries, payers and policymakers were recruited through our own networks and snowball sampling. Examples of payer organisations that were approached include healthcare insurance companies and departments of municipalities responsible for paying social care. Examples of policymakers include politicians, public servants from the ministries of health, ministries of social care, provincial or local governments and official governmental advisory bodies.

## Patient and public involvement

The selection of the outcomes in the DCE was driven by the results of eight focus groups with in total 58 persons that had multiple chronic conditions. These focus groups were organised in 2016 in the countries participating in the SELFIE project.[15] As the participants of these focus groups frequently mentioned the importance of outcomes that went beyond the traditional health outcomes such as physical functioning, we included enjoyment of life, social relationships and participation and resilience to represent aspects of well-being and 'positive health'.[16] They also frequently mentioned negative experiences related to the single-disease focus of each provider, misaligned treatment goals and advices and fragmentation or duplication of care. That is the reason why 'person-centredness' and 'continuity of care' were identified as the two key elements of experience of care.

The selection of outcomes was further informed by national workshops with stakeholder representatives in these countries, an international advisory board meeting within the SELFIE project (see acknowledgement), a review of outcomes measured in the 17 integrated care programmes and a literature review on outcome measures of integrated care.[17] In each country, two to four persons with multimorbidity and two to four informal caregivers were present at the national workshops. One patient representative and one representative from an informal caregiver organisation were present at the international advisory board meeting.

The online DCE survey was pilot tested in six elderly with multimorbidity from the Netherlands, including two think-aloud completions of the online questionnaire. Their feedback was used to improve the visual design and instructions of the questionnaire.

The results of the weight elicitation study were discussed in national stakeholder workshops and the international advisory board meetings of the SELFIE project, where both patients and informal caregivers were present. They were reimbursed for their travel and accommodation costs and did not receive other remuneration.

## Ethics approval

We adhered to the national regulations regarding medical ethics approvals and waivers. Letters of Medical Ethics Approval of the MCDA's and waivers were obtained from each country and forwarded to the European Commission as a Deliverable of the SELFIE project. All respondents read the project information and provided online consent to take part before starting the online DCE survey.

## Statistical analysis

A scale heterogeneity multinomial logit (S-MNL) model was used to analyse the responses by stakeholder group.[18 19] This model contains respondent-specific scale parameters that account for differential choice variability caused by the fact that in some respondents the error-terms are more important relative to the observed attribute-coefficients than in others. This is often interpreted as a difference in choice consistency.[20] The model also included a dummy variable to test whether respondents were more likely to choose the integrated care programme that was presented left or right. As this was not the case, this dummy was not included in the final model.

The relative importance weight of each of the included attributes was calculated as the contribution of the attribute's best level coefficient (level 3) to the sum of all attributes' best level coefficients. This was done because the weights, that will later be used in the MCDA's of the integrated care programmes,[6] need to represent the preference for the full swing from the worst to the best level of an attribute.

Differences in the mean preferences of stakeholder groups were evaluated using S-MNL models that analysed all stakeholder groups within a country simultaneously. Multiplicative interaction terms were used to assess the difference in preferences between the first stakeholder group (ie, patients with multimorbidity) and each of the other stakeholder groups. There is an indication of a difference when the 95% credible interval (CI) of the interaction term of a particular stakeholder group for a particular attribute does not contain 1.

All models were programmed in the BUGS language and estimated using OpenBUGS using Bayesian Markov Chain Monte Carlo (MCMC) methods (syntax provided in online supplemental box S2).

## RESULTS

### Respondents

Altogether, 5122 respondents completed the study, among which 1314 patients with multimorbidity. Of all patients who consented to complete the questionnaire, the proportion that completed all DCE questions was above 75%, except in Austria (67%). The mean time to complete the questionnaire was around 25 min (see online supplemental box S3).

Table 2 presents the characteristics of the respondents. Patients were older, less educated, in poorer health and

**Table 2** Characteristics of the respondents included in the analysis

| | Patients | Partners | Professionals | Payers | Policymakers |
|---|---|---|---|---|---|
| **Austria** | n=169 | n=184 | n=144 | n=99 (combined) | |
| Age mean (SD) | 51.0 (15.0) | 42.9 (14.7) | 44.7 (8.8) | 45.8 (10.7) | |
| Gender (n,% female) | 78 (46.2) | 103 (56.0) | 110 (77.5) | 53 (53.5) | |
| Educational level (n,%)* | | | | | |
| Low | 15 (8.9) | 10 (5.4) | 0 (0) | 0 (0) | |
| Medium | 126 (74.5) | 125 (67.9) | 40 (28.1) | 15 (15.2) | |
| High | 28 (16.6) | 49 (26.6) | 102 (71.8) | 81 (84.9) | |
| General health (n,%) | | | | | |
| Poor or fair | 87 (51.5) | 30 (16.3) | 4 (2.8) | 2 (2) | |
| Good | 67 (39.6) | 70 (38.0) | 33 (23.2) | 23 (23.2) | |
| Very good or excellent | 15 (8.9) | 70 (45.7) | 102 (71.8) | 24 (74.7) | |
| Number of health problems (mean, SD) | 3.4 (2.2) | 2.0 (2.0) | 0.7 (1.1) | 0.6 (1.0) | |
| **Croatia** | n=168 | n=169 | n=103 | n=73 (combined) | |
| Age | 43.1 (13.7) | 42.6 (10.8) | 44.4 (11.5) | 45.3 (10.3) | |
| Gender (n,% female) | 104 (61.9) | 96 (56.8) | 79 (76.7) | 62 (84.9) | |
| Educational level (n,%)* | | | | | |
| Low | 87 (51.8) | 91 (53.9) | 7 (6.8) | 5 (6.8) | |
| Medium | 39 (23.2) | 33 (19.5) | 38 (36.9) | 26 (35.6) | |
| High | 42 (25.0) | 45 (26.7) | 58 (56.4) | 42 (57.5) | |
| General health (n,%) | | | | | |
| Poor or fair | 74 (44.1) | 25 (14.8) | 9 (8.8) | 8 (10.9) | |
| Good | 44 (26.2) | 27 (16.0) | 26 (25.2) | 21 (28.8) | |
| Very good or excellent | 50 (29.8) | 117 (69.3) | 68 (66.0) | 44 (60.2) | |
| Number of health problems (mean, SD) | 3.2 (2.3) | 1.6 (1.6) | 1.2 (1,3) | 1.25 (1.6) | |
| **Germany** | n=160 | n=208 | n=170 | n=110 (combined) | |
| Age | 52.7 (14.6) | 46.8 (14.5) | 42.4 (10.7) | 45.6 (11.0) | |
| Gender (n,% female) | 77 (48.1) | 102 (49.0) | 123 (73.7) | 65 (59.1) | |
| Educational level (n,%)* | | | | | |
| Low | 53 (33.2) | 76 (36.5) | 31 (18.6) | 2 (1.8) | |
| Medium | 85 (53.2) | 90 (43.2) | 68 (40.7) | 36 (32.7) | |
| High | 22 (13.9) | 42 (20.2) | 68 (40.7) | 72 (65.4) | |
| General health (n,%) | | | | | |
| Poor or fair | 106 (66.3) | 49 (23.5) | 18 (10.8) | 8 (7.3) | |
| Good | 48 (30.0) | 98 (47.1) | 66 (39.5) | 36 (32.7) | |
| Very good or excellent | 6 (3.8) | 61 (29.4) | 83 (49.7) | 66 (60.0) | |
| Number of health problems (mean, SD) | 4.7 (2.5) | 2.6 (2.7) | 1.4 (1.7) | 0.87 (1.2) | |
| **Hungary** | n=192 | n=166 | n=163 | n=153 (combined) | |
| Age | 51.1 (14.1) | 43.9 (14.3) | 45.4 (10.9) | 46.7 (10.8) | |
| Gender (n,% female) | 102 (53.1) | 80 (48.2) | 84 (51.5) | 101 (66.0) | |
| Educational level (n,%)* | | | | | |
| Low | 65 (33.9) | 51 (30.7) | 1 (0.6) | 3 (2.0) | |
| Medium | 70 (36.5) | 80 (48.2) | 13 (8.0) | 21 (13.7) | |
| High | 57 (29.7) | 35 (21.1) | 149 (90.4) | 129 (84.3) | |
| General health (n,%) | | | | | |
| Poor or fair | 120 (62.5) | 44 (26.5) | 23 (14.1) | 18 (11.8) | |

**Table 2**  Continued

|  | Patients | Partners | Professionals | Payers | Policymakers |
|---|---|---|---|---|---|
| Good | 61 (31.8) | 71 (42.8) | 46 (28.2) | 50 (32.7) | |
| Very good or excellent | 11 (5.7) | 51 (30.7) | 94 (57.7) | 85 (55.6) | |
| Number of health problems (mean, SD) | 4.0 (2.4) | 2.1 (2.1) | 1.1 (1.5) | 1.1 (1.4) | |
| **Netherlands** | n=156 | n=158 | n=155 | n=104 | n=151 |
| Age | 60.4 (11.8) | 52.2 (13.0) | 42.0 (12.5) | 42.8 (11.7) | 46.3 (12.1) |
| Gender (n,% female) | 77 (49.0) | 81 (51.3) | 86 (55.5) | 48 (46.2) | 95 (62.9) |
| Educational level (n,%)* | | | | | |
| Low | 14 (9.0) | 8 (5.1) | 1 (0.6) | 0 | 2 (1.3) |
| Medium | 70 (45.0) | 76 (48.1) | 48 (31.0) | 1 (1.0) | 0 |
| High | 72 (46.2) | 74 (46.8) | 106 (68.3) | 103 (99.1) | 149 (89.7) |
| General health (n,%) | | | | | |
| Poor or fair | 86 (55.2) | 40 (25.3) | 11 (7.1) | 7 (6.7) | 14 (9.2) |
| Good | 63 (40.4) | 77 (48.7) | 59 (38.1) | 38 (36.5) | 51 (33.8) |
| Very good or excellent | 7 (4.5) | 42 (26.0) | 85 (54.9) | 59 (56.8) | 86 (57.0) |
| Number of health problems (mean, SD) | 4.0 (1.9) | 2.2 (2.5) | 1.1 (1.5) | 0.7 (1.0) | 0.8 (1.2) |
| **Norway** | n=156 | n=156 | n=162 | n=122 | n=180 |
| Age | 56.5 (14.7) | 53.7 (12.9) | 45.7 (10.7) | 51.3 (10.3) | 54.7 (13.2) |
| Gender (n,% female) | 70 (44.9) | 78 (50.00) | 124 (76.5) | 68 (55.7) | 97 (53.9) |
| Educational level (n,%)* | | | | | |
| Low | 11 (7.1) | 1 (0.6) | 2 (1.2) | 1 (0.8) | 0 |
| Medium | 52 (33.3) | 39 (25.0) | 4 (2.5) | 1 (0.8) | 35 (19.4) |
| High | 93 (59.6) | 116 (74.4) | 156 (96.3) | 120 (98.4) | 145 (80.6) |
| Poor or fair | 78 (50.0) | 38 (24.4) | 8 (4.9) | 5 (4.1) | 17 (9.4) |
| Good | 47 (30.1) | 64 (41.0) | 33 (20.4) | 17 (13.9) | 50 (27.8) |
| Very good or excellent | 31 (19.9) | 54 (34.6) | 121 (74.7) | 100 (82) | 113 (62.8) |
| Number of health problems (mean, SD) | 3.6 (1.7) | 2.2 (1.7) | 1.1 (1.5) | 0.79 (1.1) | 1.5 (1.6) |
| **Spain** | n=150 | n=151 | n=152 | | |
| Age | 62.8 (9.8) | 55.3 (11.7) | 40.8 (10.7) | | |
| Gender (n,% female) | 65 (43.3) | 103 (68.2) | 40.76 (10.7) | | |
| Educational level (n,%)* | | | | | |
| Low | 68 (45.4) | 57 (37.8) | 0 | | |
| Medium | 39 (26) | 54 (35.7) | 30 (19.7) | | |
| High | 43 (28.7) | 40 (26.4) | 122 (80.3) | | |
| General health (n,%) | | | | | |
| Poor or fair | 69 (46.0) | 45 (29.8) | 2 (1.3) | | |
| Good | 63 (42.0) | 70 (46.4) | 30 (19.7) | | |
| Very good or excellent | 18 (12.0) | 36 (23.8) | 120 (78.9) | | |
| Number of health problems (mean, SD) | 3.9 (2.5) | 2.08 (1.7) | 0.9 (1.0) | | |
| **UK** | n=163 | n=233 | n=161 | n=181 (combined) | |
| Age | 56.4 (14.4) | 45 (13.8) | 47.7 (10.8) | 45.5 (11.0) | |
| Gender (n,% female) | 94 (57.7) | 126 (54.1) | 51 (31.7) | 74 (40.9) | |
| Educational level (n,%)* | | | | | |
| Low | 60 (36.8) | 60 (25.8) | 0 | 3 (1.7) | |
| Medium | 45 (27.6) | 57 (24.5) | 2 (1.2) | 13 (7.2) | |
| High | 58 (35.6) | 116 (49.7) | 159 (98.8) | 165 (91.1) | |

**Table 2** Continued

|  | Patients | Partners | Professionals | Payers | Policymakers |
|---|---|---|---|---|---|
| General health (n,%) |  |  |  |  |  |
| Poor or fair | 129 (79.2) | 61 (26.2) | 5 (3.1) | 18 (10) | |
| Good | 28 (17.2) | 75 (32.2) | 15 (9.3) | 33 (18.2) | |
| Very good or excellent | 6 (3.7) | 97 (41.7) | 141 (87.6) | 130 (71.8) | |
| Number of health problems (mean, SD) | 4.6 (2.4) | 1.9 (2.1) | 0.7 (1.0) | 1.1 (1.6) | |

*low: no post-secondary education; high: Bachelor's degree or higher.

less often had a paid job than the other stakeholders. The health of the partners and informal caregivers was less good than the health of the professionals, payers and policymakers. There was a relatively high proportion of women among the professionals. The top five of most frequently reported diseases (not risk factors) among patients across all countries included; (1) depression, anxiety or emotional difficulties; (2) chronic back pain or sciatica; (3) diabetes; (4) stomach problem, ulcer, gastritis or reflux; and (5) cardiovascular diseases (see online supplemental box S4).

### Preferences of patients with multimorbidity

Table 3 shows the results of the regression analyses for the patients with multimorbidity. A positive regression coefficient means that respondents had a higher preference for (ie, placed a higher weight on) that level of the outcome than level one. The larger the coefficient the stronger the preference. All outcomes contributed significantly to patients' choices, in the expected direction. The DCE-coefficients of outcome-levels two and three were almost all significantly different from level one. The coefficients of level three were always higher than of level two, except for non-significant disordered levels of social relationships nd participation and person-centredness in Germany, continuity of care in Norway and Spain and costs in Spain.

The magnitude of the preference for the best level (level 3) was used as an indicator of the importance of the outcome measures. In all countries, except Hungary and Spain, enjoyment of life was valued most by patients, followed by either continuity of care (HR and DE), psychological well-being (NL and UK), physical functioning (NO) or resilience (AT). In Hungary, patients assigned the highest value to continuity of care, whereas in Spain, patients assigned the highest value to psychological well-being, both followed by enjoyment of life. In six of the eight countries (not HR and HU), physical functioning was placed in the top three. Social relationships and participation, person-centredness and costs were generally valued lower by the patients than other outcomes. The scale parameter was highest in Austria, indicating that the difference in choice consistency was greatest among the Austrian patients.

### Within-country comparison of preferences of patients and other stakeholders

Figure 2 presents the relative preference weights, sorted by importance among patients with multimorbidity. Within countries, there was considerable agreement between patients and the other stakeholders, with enjoyment of life mostly at the top and costs at the bottom. Nevertheless, there were differences.

As an indication of these differences, we counted the number of times that the 95% credibility intervals of the multiplicative interaction terms for stakeholder group and attribute excluded one (online supplemental box S5). Of all stakeholder groups, the preferences of professionals differed most often from the preferences of patients. These differences were most frequently related to physical functioning and person-centredness. Physical functioning was valued lower by professionals than patients in Austria, Germany, Netherlands, Norway and Spain. Person-centredness was valued higher by professionals than patients in Austria, Hungary, Germany and Spain.

Differences in relative weights between patients and payers/policymakers within a country were most frequently related to costs, which was more important to the latter in Austria, Germany, the Netherlands and Norway. However, the relative weight that they assigned to costs remained low. Payers and/or policymakers assigned lower values to continuity of care (HU, NL and NO), and resilience (DE, ES, HU and NL) than patients.

When the partners and other informal caregivers' weights differed from the patients, that mostly related to physical functioning, which was valued lower than patients in Austria, Germany, Netherlands and Norway and higher than patients in Spain.

### DISCUSSION
### Main findings

This is the first international preference study among such a large number of persons with multimorbidity and other stakeholders involved in decision-making about person-centred integrated care. We found that there was considerable agreement between patients, partners and other informal caregivers, professionals, payers and policymakers. In all countries, they assigned the highest or

**Table 3** Preferences for outcomes of integrated care of patients with multimorbidity

Coefficients Bayesian S-MNL model: mean and 95% credibility interval

| Outcome | Level | AT | | HR | | HU | | DE | | NL | | NO | | ES | | UK | |
|---|---|---|---|---|---|---|---|---|---|---|---|---|---|---|---|---|---|
| Physical functioning | 2 | 1.11 | 0.73–1.78 | 0.59 | 0.40–0.77 | 0.47 | 0.30–0.63 | 0.81 | 0.61–1.02 | 1.25 | 1.00–1.50 | 1.74 | 1.46–2.02 | 1.41 | 1.14–1.71 | 0.92 | 0.72–1.13 |
| | 3 | 1.19 | 0.79–1.87 | 0.71 | 0.51–0.91 | 0.69 | 0.50–0.87 | 1.00 | 0.77–1.24 | 1.95 | 1.65–2.28 | 1.93 | 1.61–2.26 | 1.44 | 1.14–1.78 | 1.22 | 0.99–1.46 |
| Psychological well-being | 2 | 0.43 | 0.22–0.71 | 0.65 | 0.47–0.82 | 0.005 | –0.14–0.16 | 0.25 | 0.07–0.43 | 0.94 | 0.73–1.17 | 0.94 | 0.72–1.16 | 1.26 | 0.97–1.59 | 0.55 | 0.37–0.73 |
| | 3 | 0.98 | 0.63–1.57 | 1.05 | 0.83–1.28 | 0.33 | 0.16–0.50 | 0.64 | 0.41–0.88 | 2.03 | 1.72–2.38 | 1.89 | 1.60–2.19 | 1.69 | 1.37–2.03 | 1.30 | 1.08–1.54 |
| Social relations and participation | 2 | 0.53 | 0.34–0.78 | 0.67 | 0.50–0.84 | 0.37 | 0.22–0.52 | 0.52 | 0.34–0.70 | 0.85 | 0.66–1.06 | 1.08 | 0.85–1.30 | 1.17 | 0.90–1.45 | 0.78 | 0.60–0.97 |
| | 3 | 0.62 | 0.39–0.97 | 0.81 | 0.61–1.01 | 0.58 | 0.41–0.75 | 0.48 | 0.30–0.68 | 1.03 | 0.81–1.27 | 1.20 | 0.97–1.44 | 1.24 | 0.98–1.54 | 1.02 | 0.83–1.22 |
| Enjoyment of life | 2 | 1.14 | 0.81–1.71 | 0.95 | 0.77–1.14 | 0.59 | 0.42–0.75 | 0.98 | 0.79–1.19 | 1.80 | 1.54–2.08 | 1.94 | 1.66–2.24 | 1.63 | 1.31–1.98 | 1.50 | 1.28–1.73 |
| | 3 | 1.56 | 1.15–2.28 | 1.41 | 1.18–1.65 | 0.72 | 0.54–0.91 | 1.36 | 1.12–1.62 | 2.76 | 2.45–3.14 | 2.68 | 2.31–3.08 | 1.65 | 1.30–2.05 | 2.16 | 1.88–2.47 |
| Resilience | 2 | 0.89 | 0.58–1.44 | 0.72 | 0.55–0.91 | 0.60 | 0.44–0.76 | 0.72 | 0.54–0.92 | 1.37 | 1.15–1.59 | 0.85 | 0.65–1.04 | 1.06 | 0.82–1.31 | 0.79 | 0.60–0.98 |
| | 3 | 1.19 | 0.85–1.80 | 1.13 | 0.93–1.34 | 0.71 | 0.54–0.89 | 1.04 | 0.83–1.27 | 1.81 | 1.56–2.08 | 1.19 | 0.97–1.42 | 1.18 | 0.92–1.47 | 1.09 | 0.88–1.30 |
| Person-centredness | 2 | 0.05 | –0.17–0.25 | 0.68 | 0.51–0.86 | 0.46 | 0.31–0.62 | 0.31 | 0.14–0.48 | 0.52 | 0.32–0.71 | 0.31 | 0.12–0.50 | 0.63 | 0.43–0.84 | 0.47 | 0.31–0.64 |
| | 3 | 0.31 | 0.11–0.53 | 0.96 | 0.78–1.15 | 0.71 | 0.53–0.90 | 0.26 | 0.08–0.44 | 0.95 | 0.74–1.16 | 0.50 | 0.31–0.70 | 0.72 | 0.50–0.95 | 0.72 | 0.53–0.90 |
| Continuity of care | 2 | 0.66 | 0.43–1.00 | 0.91 | 0.73–1.10 | 0.86 | 0.70–1.03 | 0.66 | 0.48–0.84 | 1.08 | 0.85–1.31 | 1.30 | 1.07–1.54 | 0.70 | 0.49–0.93 | 0.72 | 0.53–0.91 |
| | 3 | 0.97 | 0.70–1.38 | 1.19 | 0.99–1.40 | 1.00 | 0.82–1.18 | 1.08 | 0.85–1.32 | 1.28 | 1.05–1.53 | 1.20 | 0.97–1.44 | 0.63 | 0.42–0.85 | 0.94 | 0.74–1.15 |
| Total costs | 2 | 0.38 | 0.15–0.69 | 0.21 | 0.06–0.37 | 0.15 | –0.002–0.30 | 0.18 | 0.02–0.34 | 0.17 | –0.01–0.34 | 0.07 | –0.12–0.25 | 0.35 | 0.15–0.56 | 0.36 | 0.18–0.54 |
| | 3 | 0.44 | 0.21–0.76 | 0.24 | 0.07–0.42 | 0.19 | 0.03–0.36 | 0.27 | 0.10–0.44 | 0.43 | 0.24–0.62 | 0.23 | 0.05–0.42 | 0.15 | –0.06–0.38 | 0.58 | 0.40–0.77 |
| Scale parameter tau | | 1.39 | 0.21–1.12 | 0.57 | 0.38–0.77 | 0.72 | 0.47–1.02 | 1.07 | 0.75–1.60 | 0.65 | 0.48–0.84 | 0.58 | 0.41–0.78 | 1.23 | 0.83–2.05 | 0.56 | 0.38–0.74 |
| No. of observations | | 6120 | | 6228 | | 6984 | | 5940 | | 5760 | | 5688 | | 5400 | | 5940 | |
| No. of respondents | | 170 | | 173 | | 194 | | 166 | | 160 | | 158 | | 150 | | 165 | |

AT, Austria; DE, Germany; ES, Spain; HR, Croatia; HU, Hungary; NL, Netherlands; NO, Norway; S-MNL, scale heterogeneity multinomial logit.

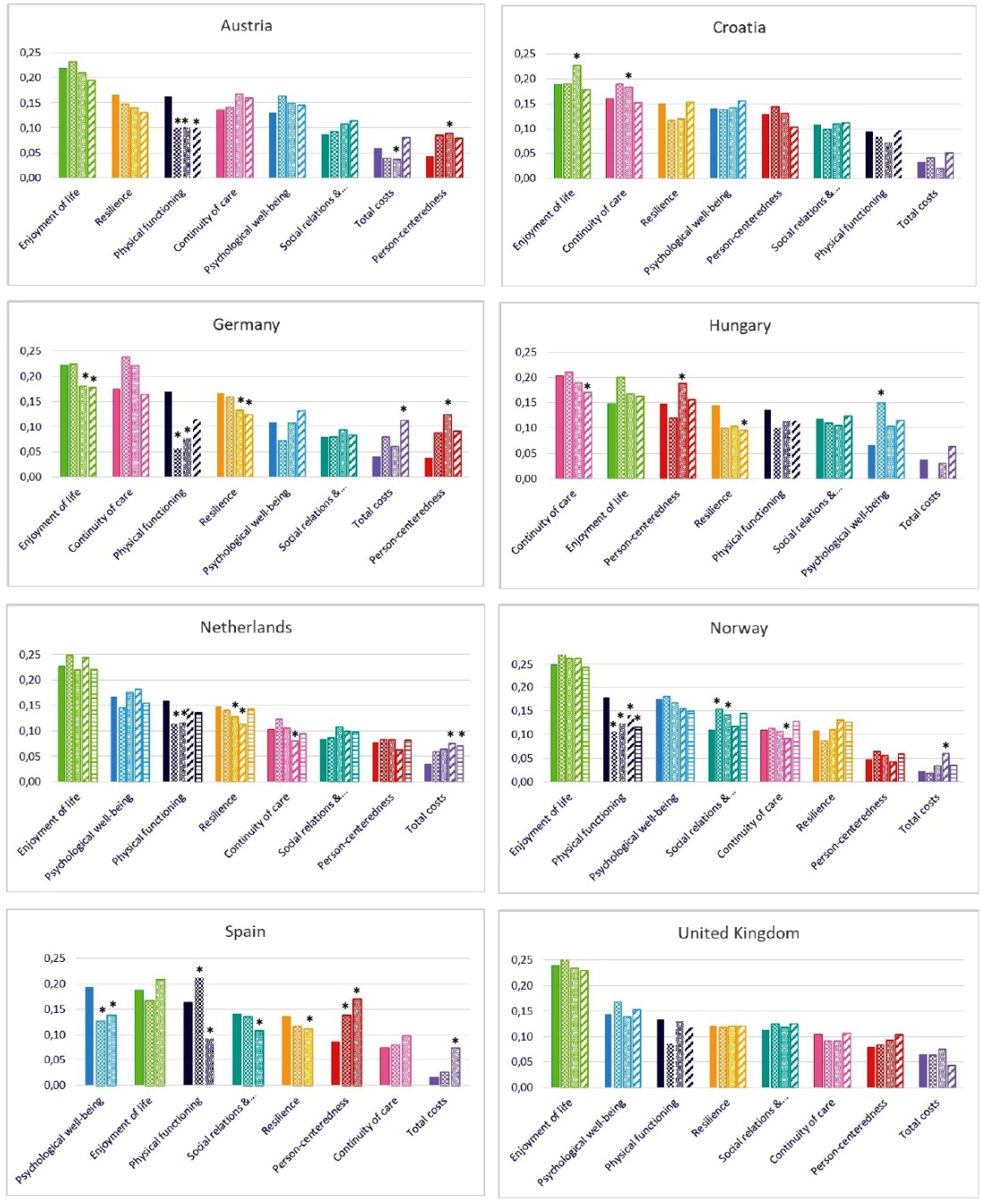

**Figure 2** Relative preferences for outcomes of all stakeholders, sorted by the preferences of patients with multimorbidity in each country (order of bars at each outcome: patients, partners and other informal caregivers, professionals, payers and policymakers). *95% credibility interval excludes 1, indicating that the stakeholder group differed from the patients with multimorbidity (see online supplemental box S5). The relative weight of the partners and other informal caregivers is 0 in Hungary. The Netherlands and Norway have five bars per attribute because they had respondents from all five stakeholder groups. The other countries combined the respondents from the payers and policymakers into one group and therefore have four bars per attribute, except for Spain which has three bars because they had not administered the questionnaire to payers and policymakers. Socialrelations &…, Social relations and participation.

second highest value to enjoyment of life and the lowest or second lowest value to costs. Of the health and well-being outcomes, psychological well-being, physical functioning and resilience were generally valued higher than social relationships and participation. Of the two experience outcomes, continuity of care was generally valued higher than person-centredness. Patients as well as the

other stakeholders in the two Eastern European countries, Hungary and Croatia, assigned a relatively high value to continuity of care. Continuity of care also entered the top-three of professionals, payers and policymakers in Germany and Austria. Representatives from Eastern European countries suggested that the high preference assigned to continuity of care might illustrate problems

with access to care and long waiting lists in various parts of their healthcare systems. German and Austrian stakeholders also pointed to a relation with capacity issues in especially the elderly care in their countries.

We also found indications of within-country differences in preferences between patients and other stakeholders. Of all stakeholder groups, professionals' preferences differed most frequently from patients' preferences and these differences most often pertained to physical functioning and person-centredness. Professionals assigned lower values to physical functioning than patients in five countries, and higher values to person-centredness in four countries. Preference heterogeneity was also found for costs, which received higher values from payers and policymakers than from patients in four countries. However, the relative value of costs compared with the other outcomes remained low for these stakeholders as well. When discussing why costs commonly ranked last, even among payers and policymakers, stakeholder representatives generally felt that this was in not in line with real world decision-making. The low value that patients assigned to costs might be related to the fact that they were informed that these costs did not have to be paid out-of-pocket. The low value assigned by payers and policymakers might be a reflection of the normative viewpoint that costs should not play an important role when it comes to health. It might also be related to the challenging task of separating their professional role from the other roles they might have. Payers and policymakers can easily relate to, or might be, patients and informal caregivers themselves.

### Comparison with previous research

In contrast to most previous DCE studies of similar interventions, we defined the DCE attributes in terms of Triple Aim outcomes and not in terms of characteristics of the services provided. The choices made by the respondents indicated that they did make trade-offs between health, well-being and experience outcomes. That finding is in line with a recent study that also measured preferences for both health and well-being outcomes in the same valuation procedure. They did so in response to the recognition that the quality-adjusted life year (QALY) does not capture the full value of interventions that involve social care.[21] Mulhern et al included the health domains of the EQ-5D (EuroQol five-dimension questionnaire) and the well-being domains of the ASCOT (Adult Social Care Outcomes Toolkit) in the same DCE and found that the top-four of most valued outcomes by the Australian general public included one ASCOT domain, namely 'control over ones daily life', which ranked fourth after the EQ-5D domains pain and discomfort, mobility and anxiety/depression.[21] They further found that social participation got the lowest value. This is in line with our study, where this outcome ended-up in the bottom-three in five countries (AT, DE, HR, HU and NL), based on patients' values.

Muhlbacher and Bethge conducted a series of online DCE's among US patients and German insurees (not only patients) that focussed on different features and building blocks of integrated service delivery.[22] In contrast to our study, costs were an important attribute in both countries. It ranked highest in Germany and in the USA only 'shared decision-making' was slightly more important. The difference is most likely explained by the fact that costs were defined as out-of-pocket costs.

Burton et al, conducted a DCE that specifically aimed to value different aspects of person-centredness in UK patients with either chronic pain or chronic respiratory disease. They found that 'taking account of the current living situation', and 'working with the patients on what they want to get from life' were the most important aspects.[23] Although our definition of person-centredness as 'care that matches an individual's needs, capabilities and preferences and where decisions are made jointly based on good information' more or less incorporated these two aspects, that did not prevent this outcome to be ranked relatively low when it was compared with the other outcomes by the patients in Western European countries.

### Limitations of the study

The first limitation is the cognitive challenge of completing a DCE. Respondents are expected to understand and interpret the definitions of the outcomes as provided. For that reason, we did not only explain the outcomes before the start of the DCE task, but also ensured that the definitions of the outcomes became visible in every choice task when respondents hovered the mouse over an outcome-heading. Respondents are further expected to consider all outcomes and make choices based on trade-offs between all outcomes. It has been reported in the literature that, although the majority of respondents does this, lower educated and less health literate respondents more often use simplified heuristics such as basing their choices on one or two high priority attributes.[24] To reduce the complexity of the choices, we have used attribute-level overlap and colour coding. Nevertheless, the proportion of patients that found the questionnaire (very) difficult varied from 8.4% in Hungary to 35.3% in Norway. However, the observation that only a small proportion of patients in all countries completed the questionnaire in less than 5 min may illustrate their engagement in the DCE task. Excluding these patients from the analyses did not affect the results (available on request).

The second limitation is related to the assumption of preference independence underlying an MCDA,[8] which means that the preference for a particular outcome is not influenced by the level of performance on the other outcomes. This was an important criterion when selecting the outcomes for the DCE. It was also the reason why enjoyment of life was framed positively (having pleasure and happiness in life) and psychological well-being was framed negatively (the absence of stress, worrying, listlessness, anxiety and feeling down). However, that does not avoid conceptual overlap entirely. We acknowledge that

enjoyment of life is a broad outcome measure that might be associated with some of the other outcomes measures.

A last limitation concerns the generalisability of the results, especially of the persons with multimorbidity and the informal caregivers which were recruited through an existing Internet panel. Because of that, respondents with lower levels of education, health literacy and health numeracy are likely to be under-represented in our study. This does not apply to Spain, where patients were recruited by professionals, which might have led to an over-representation of patients with high healthcare utilisation and informal caregivers with a higher burden of informal care.

## Implications

First, the high preference for enjoyment of life, which was defined as having pleasure and happiness in life, underlines the importance of measuring outcomes beyond health to assess the wider value of integrated care programmes for people with multimorbidity. This is not often done, and certainly not routinely. Second, this high preference for enjoyment of life calls for more attention for social and mental support, something which is not exclusively provided by the healthcare sector. Patients care more about the big picture, that is, the pleasure and happiness in life, rather than the process of care, unless there is something clearly lacking in this process, such as continuity of care. Third, although it is good that many of the current integrated care programmes already pay a lot of attention to well-being and person-centredness, these efforts should not divert attention from improving physical functioning where possible. Fourth, the differences in relative preferences for physical functioning and person-centredness between patients and professionals in several countries stress the need for shared decision-making to prioritise services that improve outcomes most important to patients. Fifth, the existence of preference heterogeneity with respect to outcomes of integrated care among stakeholders stresses the importance of informing decision-makers from multiple perspectives. The MCDA's of the 17 integrated care programmes that we are conducting are particularly suited for that.[7] These MCDA's will be used to inform pragmatic, local and regional level decisions on different options to implement, adapt or scale-up existing integrated care programmes. In this context, the downside of an MCDA of using an outcome that does not allow a comparison between integrated care and other interventions, like the QALY, is acceptable. When such a comparison is required for system-level reimbursement decisions, it can be informative to compare the results of the MCDA to the results of a cost per QALY analysis.

## CONCLUSION

Integrated care programmes aim to improve multiple outcomes. The most important outcome to patients with multimorbidity was the well-being outcome enjoyment of life, which calls for a greater involvement of the social and mental care sector. This also stresses the importance of including outcomes beyond health into the evaluation of these programmes. Process-related outcomes were less important to patients unless there were issues with the process, like the lack of continuity of care. The other stakeholders largely agreed with the preferences of the patients, although the preference heterogeneity with respect to physical functioning and person-centredness underlines the importance of shared decisionmaking between patients with multimorbidity and professionals.

**Author affiliations**
[1]Erasmus School of Health Policy & Management, Erasmus University Rotterdam, Rotterdam, The Netherlands
[2]Staff Defence Healthcare Organisation, Ministry of defence, Utrecht, The Netherlands
[3]Department of Economics, University of Bergen, Bergen, Hordaland, Norway
[4]Institut fur Hohere Studien, Wien, Austria
[5]Agency for Quality and Accreditation in Health Care and Social Welfare, Zagreb, Croatia
[6]Department of Health Care Management, Berlin University of Technology, Berlin, Germany
[7]Syreon Research Institute, Budapest, Hungary
[8]Institut d'Investigacions Biomèdiques August Pi i Sunyer (IDIBAPS), Hospital Clinic de Barcelona, Barcelona, Catalunya, Spain
[9]Centre for Health Economics, University of Manchester Institute of Population Health, Manchester, UK

**Acknowledgements** We would like to thank all the anonymous respondents who devoted time to the completion of the weight-elicitation study. We are grateful for the contributions of the members of the international advisory board of the SELFIE project and the participants of the national workshops. The advisory board consisted of Rudi Westendorp (Center for Healthy Ageing), Ronald van Breugel (health insurer VGZ), Jan de Maeseneer (Emeritus professor of Family Medicine), Mieke Rijken (NIVEL), Viktoria Stein (International Foundation of Integrated Care), Luke Slawomirski (OECD), Ian Forde (formerly OECD), Rick Greene (International Alliance of Carer Organizations), Nadine Henningsen (Canadian Homecare Organization), Dominik Tomek, (European Patients' Forum), Derek Steward (NHS NIHR, Patient and Public Health Involvement), Antonia Croy (patient advocate) and Ikka Kunnamo (Editor in Chief Evidence-Based Medicine Guidelines, Duodecim Medical Publications). We are also grateful for the input that we have received from the participants of the national workshops in the eight participating countries.

**Contributors** MRvM was the coordinator of SELFIE, designed the study, had access to the data from all countries, supervised the analyses, controlled the decision to publish and wrote the paper. She accepts full responsibility for the conduct of the study. MK was part of the coordinating team of SELFIE, contributed to data analysis, interpretation of data for all countries and writing the paper. FL was part of the coordinating team of SELFIE, co-designed the study, contributed to data collection, data analyses, interpretation of data for all countries and reviewed the paper. MH was part of the coordinating team of SELFIE, co-designed the study, contributed to data collection, data analyses, interpretation of data for all countries and reviewed the paper. WL was part of the coordinating team of SELFIE, co-designed the study, contributed to data collection, data analyses, interpretation of data for all countries and reviewed the paper. KI contributed to the translation of the DE into Norwegian, data collection and interpretation in Norway and reviewed the paper. JEA contributed to the translation of the DE for Norway, data collection and interpretation in Norway and reviewed the paper. MK contributed to the translation of the DE for Austria, data collection and interpretation in Austria and reviewed the paper. VS contributed to the translation of the DE for Germany, data collection and interpretation in Germany and reviewed the paper. DR contributed to the translation of the DE for Croatia, data collection and interpretation in Croatia and reviewed the paper. JP contributed to the translation of the DE for Hungary, data collection and interpretation in Hungary and reviewed the paper. IC contributed to the translation of the DE for Spain, data collection and interpretation in Spain and reviewed the paper. JS contributed to the translation of the DE for the UK, data collection and interpretation in the UK and reviewed the paper. MJ optimised the DE designs, conducted the Bayesian statistical analyses and reviewed the paper.

**Funding** The SELFIE project has received funding from the European Union's Horizon 2020 research and innovation programme under grant agreement No 634288.

**Disclaimer** The content of this paper reflects only the SELFIE group's views and the European Commission is not liable for any use that may be made of the information contained herein.

**Competing interests** None declared.

**Patient consent for publication** Not required.

**Provenance and peer review** Not commissioned; externally peer reviewed.

**Data availability statement** No data are available. The individual-level DCE data will not be made publicly available. Upon request to the corresponding author, we may consider in collaboration with the respective SELFIE-partner, whether country-specific data can be shared. Other findings from the overarching SELFIE study can be found on the SELFIE website https://www.selfie2020.eu/.

**ORCID iD**
Maureen Rutten-van Mölken http://orcid.org/0000-0001-8706-3159

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
