## [Reviewer comments · BMJ Open]

ARTICLE DETAILS

TITLE (PROVISIONAL)	Comparing patients' and other stakeholders' preferences for outcomes of integrated care for multimorbidity: a discrete choice experiment in eight European countries.
AUTHORS	Rutten-van Molken, Maureen; Karimi, Milad; Leijten, Fenna; Hoedemakers, Maaïke; Looman, Willemijn; Islam, Kamrul; Askildsen, Jan; Kraus, Markus; Ercevic, Darija; Struckmann, Verena; Pitter, Janos; Cano, Isaac; Stokes, Jonathan; Jonker, Marcel

VERSION 1 – REVIEW

REVIEWER	Cathleen Muche-Borowski University Medical Center Hamburg-Eppendorf Institute and polyclinic for general medicine, Center for Psychosocial Medicine Martinistrasse 52 20246 Hamburg Germany
REVIEW RETURNED	26-Feb-2020

GENERAL COMMENTS	This concerns the result part: page 9, lines 19-31: Please check the country information and compare this with Table 3. Example: why is Spain mentioned as an exception for enjoyment of life with a value of 1.65 (1.30-2.05), for HU this is understandable with a value of 0.72 (0.54-0.91) Page 9, lines 42-54: Please adapt the order of countries in the text in which the countries are listed in the table. Where does the reader find the information about the costs in Box S5 for P4 / 5 vs P1 for ES, NL or NO, but which are mentioned in the text lines 53-54. Page 10, lines 4-5: Where does the reader find the information for P4/5 vs P1 for continuity of care for ES, NL or NO, which are described in the text. Please check the explanations in the results and the discussion accordingly. Fig.2 Please insert a legend of coloured bars, sometimes there are 4 or 5 bars.
---

	It would be advantageous: the same order of the information on the relative preferences and thus the same order of the colored bars.
--	--

REVIEWER	Dorijn Hertroijds Maastricht University
REVIEW RETURNED	20-Apr-2020

GENERAL COMMENTS	The authors conducted a discrete choice experiment to elicit preferences of patients, professionals, partners, policy makers and payers for outcomes of integrated care for multimorbidity across several European countries. The study is well-performed, the article is clearly written and the figures and tables are clear and visually attractive. The article was very interesting to read. I only have a few minor comments.  1. In the abstract under 'main outcome measures' you wrote that the main outcome measures are the attributes you included in the DCE. However, the main outcome measures are the preferences of the participants regarding outcomes of integrated care. 2. Introduction, pg 4, line 54: add a comma before the word 'because'. 3. While reading the method section, I was wondering how you determined the attributes and levels included in the DCE. There is no mention of this under 'DCE design' or 'questionnaire design', which is where I would have expected it. Towards the end of the section you explain the determination of the attributes well, but I feel that you could have explained this earlier. It remains unclear how you determined the levels of the attributes. 4. Your study is well executed. Therefore, in the discussion, would you not want to add the strength of your study? 5. When it comes to DCEs, I'm often interested in the mean relative importance of each attribute. I think this provides a clearer overview of the most important attribute than the magnitude of the preference for the best level.
--

VERSION 1 – AUTHOR RESPONSE

Reviewer(s)' Comments to Author:

Reviewer: 1

Reviewer Name: Cathleen Muche-Borowski

Institution and Country: University Medical Center Hamburg-Eppendorf Institute and polyclinic for general medicine, Center for Psychosocial Medicine Martinistrasse 52
20246 Hamburg

Germany

Please state any competing interests or state 'None declared': Non declared

We would like to thank this reviewer for her time and efforts to review our paper.

Please leave your comments for the authors below This concerns the result part:
page 9, lines 19-31:

Please check the country information and compare this with Table 3.

Example: why is Spain mentioned as an exception for enjoyment of life with a value of 1.65 (1.30-2.05), for HU this is understandable with a value of 0.72 (0.54-0.91)

Reply: The numbers in Table 3 are correct; so are the numbers in the sentence in the text. In Spain enjoyment of life ranks second (coefficient 1.65), just after psychological wellbeing (coefficient 1.69), although the difference is small. This is based on the coefficients of the best levels, as explained in the methods. Therefore, row 23-25 reads "In Hungary, patients assigned the highest value to continuity of care, whereas in Spain, patients assigned the highest value to psychological wellbeing, both followed by enjoyment of life." Table 3 was entirely checked but no errors found.

Page 9, lines 42-54:

Please adapt the order of countries in the text in which the countries are listed in the table.

Where does the reader find the information about the costs in Box S5 for P4 / 5 vs P1 for ES, NL or NO, but which are mentioned in the text lines 53-54.

Reply: In line 54 Hungary and Germany switched order to ensure the order is the same as in the table. The information about the costs can be found in Box S5. For example, in the column 'P4 vs P1' for NL, the coefficient of the interaction term for total cost is 1.53 with a 95% CI of 1.19 to 1.93. Because the CI excludes 1, the difference between P4 (payers) and P1 (patients with multimorbidity) is significant.

Page 10, lines 4-5:

Where does the reader find the information for P4/5 vs P1 for continuity of care for ES, NL or NO, which are described in the text.

Please check the explanations in the results and the discussion accordingly.

Reply: The information about the continuity of care can be found in Box S5. For example, in the column 'P4 vs P1' for NL, the coefficient of the interaction term for continuity of care is 0.73 with a 95% CI of 0.56 to 0.91. Because the CI excludes 1, the difference between P4 (payers) and P1 (patients with multimorbidity) is significant.

Fig.2

Please insert a legend of coloured bars, sometimes there are 4 or 5 bars.

It would be advantageous: the same order of the information on the relative preferences and thus the same order of the colored bars.

Reply: Might it be that the reviewer has missed the title and footnote below Fig 2, because it was put on a separate page. It states "Figure 2. Relative preferences for outcomes of all stakeholders, sorted by the preferences of patients with multi-morbidity in each country (order of bars at each outcome: patients, partners and other informal caregivers, professionals, payers and policy makers)" Hence, the order of the bars was explained.

In all countries except the Netherlands and Norway, the payers and policy makers were combined into one group. That is why the Netherlands and Norway have five bars per attribute and the other countries four. We added the following information to the footnote. "The Netherlands and Norway have 5 bars per attribute because they had respondents from all five stakeholder groups. The other countries combined the respondents from the payers and policy makers into one group and therefore have 4 bars per attribute, except for Spain which has 3 bars because they had not administered the questionnaire to payers and policy makers."

We have deliberately not used same order of the colored bars for each country but used one color per attribute, so that the reader can immediately see for example that enjoyment of life, which is green in all countries, ranks second in Hungary and Spain.

Reviewer: 2

Reviewer Name: Dorijn Hertroijs

Institution and Country: Maastricht University Please state any competing interests or state 'None declared': None declared

Please leave your comments for the authors below The authors conducted a discrete choice experiment to elicit preferences of patients, professionals, partners, policy makers and payers for outcomes of integrated care for multimorbidity across several European countries. The study is well-performed, the article is clearly written and the figures and tables are clear and visually attractive. The article was very interesting to read.

We would like to thank this reviewer for her time and efforts to review our paper and for her compliments.

I only have a few minor comments.

1. In the abstract under 'main outcome measures' you wrote that the main outcome measures are the attributes you included in the DCE. However, the main outcome measures are the preferences of the participants regarding outcomes of integrated care.

Reply: This was changed exactly as suggested by the reviewer.

2. Introduction, pg 4, line 54: add a comma before the word 'because'.

Reply: comma added

3. While reading the method section, I was wondering how you determined the attributes and levels included in the DCE. There is no mention of this under 'DCE design' or 'questionnaire design', which is where I would have expected it. Towards the end of the section you explain the determination of the attributes well, but I feel that you could have explained this earlier. It remains unclear how you determined the levels of the attributes.

Reply: How the attributes were determined is explained in the methods section under subsection 'Patient and public involvement' (an obligatory subsection in the BMJ) because the selection of the outcome measures was based on focus groups. These focus groups are further explained there. To mention this earlier in the methods section, we now added "The outcomes were largely based on focus groups" to the seventh line of the methods section.

Because the weights will later be used in the Multi-Criteria Decision Analyses (MCDA's) of the integrated care programmes in the SELFIE project they need to represent the preference for the full swing from the worst to the best level of an attribute. That guided the definition of the best and worst level. The wording of these levels was chosen in line with that (e.g. 1. always or mostly, 2. regularly, 3. seldom or never; 1. no or barely, 2. some, 3. a lot; 1. poor, 2. fair, 3. good). How the cost-levels are determined is explained on page 5, i.e. "The three levels of costs were based on country-specific estimates of the mean total health and social care costs for people with multimorbidity in 2017 (middle level) and increased and decreased by 20% to obtain the poor and good performance level. This percentage was chosen to ensure enough variation in levels while at the same time remaining within realistic boundaries."

4. Your study is well executed. Therefore, in the discussion, would you not want to add the strength of your study? "

Reply: Thank you for this suggestion, but we feel that the strength was highlighted in the first sentence of the discussion which reads "This is the first international preference study among such a large number of persons with multimorbidity and other stakeholders involved in decision making about person-centered integrated care." It was also highlighted in the stand-alone strengths and limitation box which includes:

- We designed an online DCE to measure the relative importance of health, well-being, experience, and cost outcomes of integrated care to patients with multimorbidity from eight European countries.***

- *The choice of outcomes was largely driven by focus groups involving patients with multimorbidity.*
- *We compared patients' preferences with the preferences of other stakeholders including informal caregivers, professionals, payers, and policy makers in a large sample of 5122 respondents.*

5. When it comes to DCEs, I'm often interested in the mean relative importance of each attribute. I think this provides a clearer overview of the most important attribute than the magnitude of the preference for the best level.

Reply: The reason that we concentrate on the relative importance-weight of the best level of each attribute is that these weights will later be used in the Multi-Criteria Decision Analyses (MCDA's) of the integrated care programmes in the SELFIE project. For that purpose, the weights need to represent the preference for the full swing from the worst to the best level of an attribute. As a consequence we cannot use the mean importance. However, to inform the reviewer we have put the mean relative importance next to the best-level relative importance in the table below. The table shows that the differences are small.

Comparison of best level importance and mean level importance by country

	AU		DE		ES		HR	
	Level 3	Level2+ 3	Level 3	Level2+ 3	Level 3	Level2+ 3	Level 3	Level2+ 3
Physical functioning	0,16	0,16	0,17	0,15	0,16	0,16	0,09	0,10
Psychological well-being	0,13	0,11	0,11	0,08	0,19	0,16	0,14	0,13
Social relations & part	0,09	0,10	0,08	0,10	0,14	0,14	0,11	0,11
Enjoyment of life	0,22	0,21	0,22	0,21	0,19	0,20	0,19	0,18
Resilience	0,17	0,16	0,17	0,15	0,14	0,13	0,15	0,14
Person-centeredness	0,04	0,06	0,04	0,07	0,09	0,09	0,13	0,13
Continuity of care	0,14	0,14	0,17	0,17	0,07	0,08	0,16	0,16
Total costs	0,06	0,06	0,04	0,06	0,02	0,04	0,03	0,03

	HU		NL		NO		UK	
	Level 3	Level2+ 3	Level 3	Level2+ 3	Level 3	Level2+ 3	Level 3	Level2+ 3
Physical functioning	0,14	0,13	0,16	0,15	0,18	0,15	0,13	0,14
Psychological well-being	0,07	0,06	0,17	0,14	0,17	0,14	0,14	0,12
Social relations & part	0,12	0,12	0,08	0,09	0,11	0,09	0,11	0,12
Enjoyment of life	0,15	0,15	0,23	0,22	0,25	0,22	0,24	0,24
Resilience	0,14	0,15	0,15	0,15	0,11	0,15	0,12	0,13
Person-centeredness	0,15	0,14	0,08	0,08	0,05	0,08	0,08	0,08
Continuity of care	0,20	0,21	0,10	0,12	0,11	0,12	0,10	0,11
Total costs	0,04	0,04	0,03	0,04	0,02	0,04	0,06	0,06

VERSION 2 – REVIEW

REVIEWER	Dr. Cathleen Muche-Borowski, MPH Department of General Practice and Primary Care University Medical Center Hamburg-Eppendorf Hamburg, Germany
REVIEW RETURNED	28-Jun-2020

GENERAL COMMENTS	in this version i can't find figure 1 and 2. I had see these figures in former versions of your manuscript.
---

REVIEWER	Dorijn Hertroijs Maastricht University, the Netherlands
REVIEW RETURNED	15-Jul-2020

GENERAL COMMENTS	As said before, this is a well executed study, which requires no further improvements. I did find one spelling error on page 10, line 46: ".this was in not in line.
--